# Phenol Degradation by *Pseudarthrobacter phenanthrenivorans* Sphe3

**DOI:** 10.3390/microorganisms11020524

**Published:** 2023-02-18

**Authors:** Stamatia Asimakoula, Orfeas Marinakos, Epameinondas Tsagogiannis, Anna-Irini Koukkou

**Affiliations:** Laboratory of Biochemistry, Sector of Organic Chemistry and Biochemistry, Department of Chemistry, University of Ioannina, 45110 Ioannina, Greece

**Keywords:** phenol, biodegradation, *Pseudarthrobacter phenanthrenivorans* Sphe3, *ortho*-cleavage pathway, phenol hydroxylase, catechol dioxygenase, *cis*, *cis*-muconate, alginate immobilization

## Abstract

Phenol poses a threat as one of the most important industrial environmental pollutants that must be removed before disposal. Biodegradation is a cost-effective and environmentally friendly approach for phenol removal. This work aimed at studying phenol degradation by *Pseudarthrobacter phenanthrenivorans* Sphe3 cells and also, investigating the pathway used by the bacterium for phenol catabolism. Moreover, alginate-immobilized Sphe3 cells were studied in terms of phenol degradation efficiency compared to free cells. Sphe3 was found to be capable of growing in the presence of phenol as the sole source of carbon and energy, at concentrations up to 1500 mg/L. According to qPCR findings, both pathways of *ortho*- and *meta*-cleavage of catechol are active, however, enzymatic assays and intermediate products identification support the predominance of the *ortho*-metabolic pathway for phenol degradation. Alginate-entrapped Sphe3 cells completely degraded 1000 mg/L phenol after 192 h, even though phenol catabolism proceeds slower in the first 24 h compared to free cells. Immobilized Sphe3 cells retain phenol-degrading capacity even after 30 days of storage and also can be reused for at least five cycles retaining more than 75% of the original phenol-catabolizing capacity.

## 1. Introduction

Phenolic compounds are considered major environmental pollutants originating from numerous industrial processes such as manufacturing of pulp and paper, dyes and textiles, fertilizers and pesticides, processing of leather and coal, as well as in steel and oil refineries and pharmaceutical industries [1,2,3,4,5]. These compounds are considered to be major toxic pollutants, even at low concentrations, posing an ecological threat and endangering human life through their bioaccumulation in the environment [1,6]. Furthermore, phenolic compounds can deter plant and animal growth, while acute or chronic exposure to them can adversely affect human health, with serious impacts on the body’s systems and functions, including the nervous, immune, and respiratory systems, causing several deficiencies and occasionally leading even to cancer [4].

There are several methods to remove phenol from wastewater/effluents/the environment including solvent extraction, adsorption, chemical oxidation, and biodegradation [7,8], with the latter being the most efficient one considering the cost, low energy consumption, and not creating secondary by-products that can act as pollutants themselves. Bioaugmentation enhances natural biodegradation and the immobilization of microbial cells with degrading capability on a carrier material is widely used in bioremediation.

Cell immobilization prevails over using free bacterial cells for bioremediation applications, solving problems such as substrate inhibition, sensitivity to environmental factors, as well as settling issues with recovery and reusability [9]. Specifically, the use of gel polymers in bacterial immobilization has proved to be more efficient in terms of phenol toxicity and degradation rates [10,11].

A number of studies report bacteria that are capable of metabolizing phenol, which belong to genera *Pseudomonas* [10,12,13], *Acinetobacter* [2,14], *Rhodococcus* [3,15], *Bacillus* [16], and *Arthrobacter* [17,18,19,20].

Bacterial aerobic degradation of phenol proceeds through the dihydroxylation of its benzene ring by the enzyme phenol hydroxylase, resulting in the formation of catechol [21]. Catechol is further degraded via either the *ortho*- or *meta*-pathway. In the *ortho*-pathway (also known as the *β*-ketoadipate pathway), the bond between the hydroxyl groups of catechol is cleaved by catechol 1,2- dioxygenase resulting in the formation of *cis*,*cis*-muconate. In the *meta*-cleavage pathway, catechol is transformed to 2-hydroxymuconic semialdehyde by a ring cleavage at the bond adjacent to the hydroxyl groups by catechol 2,3-dioxygenase. The products from both pathways are further metabolized into molecules that fuel the Krebs cycle [22,23].

*Pseudarthrobacter phenanthrenivorans* Sphe3 is a Gram-positive bacterium, isolated from a creosote-polluted site in Epirus and it possesses the ability to catabolize polycyclic aromatic hydrocarbons (PAHs) [24]. In the present study, we successfully cultivated *P. phenanthrenivorans* Sphe3 in the presence of various concentrations of phenol, as the sole source of carbon and energy. In silico studies on the Sphe3 genome revealed genes implicated in phenol catabolism, possibly coding for phenol hydroxylase, 1,2- and 2,3-catechol dioxygenase. Hence, in order to shed light on the Sphe3 phenol catabolism pathways, the transcription levels of the aforementioned genes were assessed when the strain was grown in the presence of phenol. Sphe3 cells were entrapped into alginate beads and were used to monitor phenol catabolism in comparison with free cells. In addition, the reusability and the storage stability of the immobilized cells were tested.

## 2. Materials and Methods

### 2.1. Bacterial Strain and Growth Conditions

The bacterial strain *Pseudarthrobacter phenanthrenivorans* Sphe3 used in the present study was isolated from a creosote-polluted area in Epirus, Greece [24]. Sphe3 was grown in lysogeny broth (LB) medium or in minimal medium M9 (MM M9), prepared as described before [25], in the presence of phenol (300–2000 mg/L) and glucose (400 mg/L) as the sole carbon and energy sources, on a rotary shaker agitated at 180 rpm, at 30 °C.

### 2.2. Assessments of Growth and Determination of Residual Phenol in Culture Medium

*P. phenanthrenivorans* cells were grown towards the mid of log phase in LB medium incubated at 30 °C under agitation. The cells were centrifuged, washed, appropriately diluted, and resuspended in MM M9 with different phenol concentrations as the sole carbon source. The cultures were inoculated at a starting O.D._600nm_ of 0.15 and bacterial growth was monitored by measuring the optical density at 600 nm at various time points. Cultures inoculated with boiled dead cells were used in parallel as the abiotic negative controls. To determine the viable cell number, the spread plate technique was used to enumerate colony forming units (CFUs). Specifically, 100 μL of the Sphe3 culture were used for plating onto LB agar plates and the viable counts were established after 48 h incubation at 30 °C.

To monitor phenol removal, 1 mL samples were taken in regular intervals, centrifuged, filtered with 0.22 μm filters, and phenol concentration was measured using the 4-aminoantipyrine colorimetric method [26]. The reaction mixture was prepared by mixing 2% 4-aminoantipyrine solution, 8% potassium ferricyanide solution, and 2 N ammonium hydroxide solution with a ratio of 1:1:2, respectively. The pH was adjusted to be 6.9 ± 0.1 using a pH meter. Then, the sample was diluted 1:100 with distilled water and 0.01 mL were added to 0.1 mL of the reaction mixture. This mixture was diluted again with 1 mL distilled water and allowed to react for 15 min at room temperature. The absorbance was then measured at 510 nm using the Shimadzu UV-1201 spectrophotometer. The phenol concentration was elicited from a calibrated standard curve. All determinations were made in triplicate.

### 2.3. Preparation of Sphe3 Cell Extracts

Sphe3 cells were grown in phenol (500 mg/L) and harvested at the mid-exponential phase of the growth curve by centrifugation at 6000× *g* for 15 min at 4 °C, and were subsequently washed with 50 mM Tris-HCl buffer (pH 8) containing 1 mM dithiothreitol (DTT) and 10 mM phenylmethylsulfonyl fluoride (PMSF), resuspended in 2 mL of the same buffer and disrupted with a mini Bead Beater (Biospec Product, Bartlesville, OK, USA) (10 times of 1 min periods using zirconium beads 0.1 mm in diameter). The homogenate was centrifuged at 12,000× *g* for 20 min at 4 °C and the collected supernatant was kept on ice to prevent the inactivation of the enzymes and was further used as the cell-free extract for the enzyme assays. Protein concentration in the crude enzyme was determined spectrophotometrically by applying the Bradford method using the Bio-Rad reagent (Bio-Rad Laboratories, Hercules, CA, USA) and bovine serum albumin (BSA) (Amresco Inc., Solon, OH, USA) as the standard [27].

### 2.4. Enzyme Assays

Phenol hydroxylase activity in Sphe3 cell extracts was determined by the oxidation of NADH in the presence of phenol, measuring the absorbance at 340 nm. The 1 mL reaction contained 50 mM Tris-HCl buffer (pH 8), 50 μL cell extract, and 1 mM phenol. One unit of phenol hydroxylase activity was defined as the amount of enzyme catalyzing the oxidation of 1 μmol NADH min^−1^ [28].

Catechol dioxygenases activity was assayed in a 1 mL reaction mixture containing 50 mM Tris-HCl buffer (pH 8), 50 μL cell extract, and 3 mM catechol. Each reaction began with the addition of the substrate. Buffer containing only the enzyme and buffer containing only the substrate were used as controls. Catechol 1,2-dioxygenase activity was measured spectrophotometrically as an increase in absorbance at 260 nm by following the formation of *cis*-*cis* muconic acid, the *ortho*-cleavage product of catechol [29]. Catechol 2,3-dioxygenase activity was measured spectrophotometrically as an increase in absorbance at 375 nm by following the formation of 2-hydroxymuconic semialdehyde, the *meta*–cleavage product of catechol [30]. The enzyme activities were expressed as moles of product formed per min per mg of protein. The molar extinction coefficients 16,800 mM^−1^ cm^−1^ (muconic acid) and 14,700 mM^−1^ cm^−1^ (2-hydroxysemialdehyde) were used to determine the activities for catechol 1,2-dioxygenase and catechol 2,3-dioxygenase, respectively [29,30].

All enzyme activity assays were performed in triplicates, in quartz cuvettes at 25 °C using a Shimadzu UV-1201 spectrophotometer (Triad Scientific, Inc., Manasquan, NJ, USA).

### 2.5. Quantitative Real-Time PCR (RT-qPCR)

Total RNA isolation was performed using the NucleoSpin^®^ RNA Isolation kit by MACHEREY-NAGEL (Düren, Germany) according to manufacturer’s instructions with slight modifications on the sample homogenization procedure. Specifically, Sphe3 cells were grown in MM M9 supplemented with 500 mg/L phenol as the sole source of carbon and energy. The cell pellet was obtained by harvesting Sphe3 culture at the mid-exponential phase of growth and centrifuging at 6000× *g* for 15 min at 4 °C. Then, the cell pellet was resuspended in 100 μL TE buffer (10 mM Tris-HCl, 1 mM EDTA; pH 8) containing 10 mg/mL lysozyme by vigorous vortexing. The resulting solution was incubated for 1 h at 37 °C.

cDNA synthesis was performed using the PrimeScript™ RT Reagent Kit with gDNA Eraser (Perfect Real Time, Takara Bio Inc., Kusatsu, Shiga, Japan) according to the manufacturer’s instructions and stored at −20 °C. cDNA was further diluted and used as a template at a final concentration of 2.5 ng. The expression levels of the target genes were quantified by RT-qPCR in the CFX Connect Real-Time PCR Detection System (Bio-Rad, United States) using the Kapa SYBR Fast qPCR Kit Master Mix (2×) Universal (Kapa Biosystems, Wilmington, MA, USA), performed as described previously [31].

Primers used in the present study are listed in Appendix A. The efficiency (E) of one cycle RT-qPCR in the exponential phase was found to be 1.89–2.05 (E = 10^(−1/slope)^) [32,33] with correlation factors 0.9918 < R^2^ < 0.9959. The housekeeping gene *gyr*β was used as the reference gene and gene expression levels in glucose were used as a calibrator. The results were analyzed by the relative quantification method [34].

### 2.6. Immobilization of Sphe3 Cells in Calcium Alginate Beads

Sphe3 immobilization was performed as described before [35] with slight modifications. In short, a mid-exponentially grown culture of Sphe3 in LB medium was collected and centrifuged. The cell pellet obtained corresponded to 3 × 10^8^ CFUs according to the spread plate technique. Thereafter, the cell pellet was washed three times with 0.9% *w*/*v* sterile NaCl solution and resuspended in 10 mL of 4% *w*/*v* sodium alginate solution (prior to being autoclaved at 121 °C for 20 min). The mixture was then dropped with an insulin syringe into a gently stirred 3% *w*/*v* filtered solution of CaCl_2_ at a volumetric ratio of 1:5. Calcium alginate gel beads were spontaneously formed in this cross-linking reaction and left overnight at 4 °C under stirring. After stabilization, the beads were washed with 0.9% *w*/*v* NaCl three times and were ready for further use.

### 2.7. Phenol Degradation by Immobilized Sphe3 Cells

The immobilized Sphe3 cells in sodium alginate beads were transferred in MM M9 without phosphates, to avoid alginate bead dissolution [36], containing 1000 mg/L phenol and incubated at 30 °C. MM M9 without phosphates containing phenol and lacking beads was used as a control. Samples were withdrawn at various time points, centrifuged at 11,000× *g* for 10 min, and filtered with 0.22 μm filters. Phenol removal was determined as described above. All determinations were made by triplicate.

### 2.8. Phenol Catabolism—Parameters Optimization

The effects of pH (4–8) and temperature (20–50 °C) on phenol catabolism in Sphe3 cultures for both free and immobilized cells were investigated. Each experiment was conducted by keeping all but the tested parameters constant. Measurements to estimate the percentage removal of phenol were taken after incubation with phenol for 48 h.

### 2.9. Reusability and Phenol Degradation Efficiency by Immobilized Cells after 30 Days of Storage

The reusability of the immobilized cells was determined through phenol concentration measurements. Each reaction cycle was carried out for 48 h. At the end of each batch, beads were collected by filtration, rinsed with sterile 0.9% *w*/*v* NaCl solution, and then added to the next batch. The process was repeated for five successive cycles. The activity of the immobilized cells after each cycle against phenol was assessed by means of residual phenol. The activity of the first cycle was considered as 100%.

Alginate beads with entrapped Sphe3 were stored at 4 °C in 0.2 M CaCl_2_ for a period of 30 days. In order to check the immobilized cells’ phenol-degrading capacity at 10, 20, and 30 days of storage, alginate beads were rinsed with sterile 0.9% *w*/*v* NaCl solution, placed in a fresh medium containing phenol, and incubated as described above. Samples were taken after 48 h of incubation to determine the residual phenol concentration and assess phenol degradation efficiency.

### 2.10. High-Performance Liquid Chromatography (HPLC) Analysis

For further confirmation of the phenol biodegradation, the cell-free supernatants, collected as described above, were eluted with methanol (1:1 ratio), filtered with 0.45 μm filters, and used to determine the phenol concentration by high-performance liquid chromatography (HPLC) (Shimadzu, LC-10A, Tokyo, Japan) using a Bondapack C18 column, particle size 10 μm, length 300 mm, diameter 3.9 mm, and a diode array UV detector as described elsewhere [37]. The mobile phase consisted of acetonitrile (A) and 0.1% acetic acid in water (B). The elution conditions applied for solvent B were as follows: 0–30 min 80–50%, 30–35 min 50%, and 35–40 min 80%. Elution was performed at 27 °C with a flow rate of 1 mL min−1, and the samples were detected at 280 nm. The retention times of phenol and catechol were 6.4 min and 4.7 min, respectively. The quantification and characterization of phenol, catechol, and *cis*,*cis*-muconate (*cc*MA) were based on standard compounds and calibration curves under the same conditions.

### 2.11. Statistical Analysis

All measurements were carried out in triplicates and each result was expressed as mean ± standard deviation (SD). Statistical analysis was performed by two-way analysis of variance (ANOVA) using GraphPad Prism 9.0.0 software package and statistical significance of the data was evaluated considering *p* < 0.05. Multiple comparisons were performed using Tukey’s test.

## 3. Results and Discussion

### 3.1. Utilization of Phenol as the Sole Source of Carbon and Energy

The ability of several bacterial strains to utilize phenol either aerobically or anaerobically has been reported by many scientists [38,39]. Aerobic processes are characterized by lower cost and higher degradation efficiency due to microorganisms’ rapid growth and ability to completely mineralize the xenobiotics [40]. For instance, some characteristic strains that utilize the aerobic catabolism of phenols belong to the genera of *Acinetobacter*, *Pseudomonas*, *Rhodococcus*, and *Kocuria* [2,3,10,41] while phenol-degrading strains that belong to the *Arthrobacter* genus are particularly few [17,19,20,42,43].

*Pseudarthrobacter phenanthrenivorans* Sphe3 can grow in the presence of various aromatic compounds, as reported before [24,31]. In this study, Sphe3 was found capable of growing on phenol as the sole carbon and energy source, at concentrations of up to 1500 mg/L (Figure 1), while no growth was observed at 2000 mg/L phenol.

The growth curves showed no lag phase and Sphe3 cells grew exponentially between 0 and 12 h in the presence of up to 1000 mg/L phenol in MM, whereas at higher concentrations (1200 or 1500 mg/L) a delayed growth was observed. Lower growth at 300 mg/L phenol could be explained by the growth-limiting factor of the substrate (carbon source) concentration, while the lower growth at phenol concentration above 1000 mg/L suggest the toxic effect of phenol on Sphe3 cells. A similar phenomenon of increasing growth and degradation efficiency with increasing the substrate concentration and then decreasing above a concentration threshold, indicating substrate inhibition kinetics has been reported by others [2,44,45].

Generally, there is a relationship between cell growth and phenol degradation [5]. As expected, the growth is associated with phenol catabolism by Sphe3 cells. Increases in bacterial growth, as depicted in Figure 1, was correlated with decreases in phenol concentration, indicating that Sphe3 cells utilize phenol as the sole carbon source. Figure 2 depicts the phenol catabolism of Sphe3, estimated by measuring the remaining phenol in the culture medium by the 4-AAP method. Specifically, the highest phenol removal was observed after incubating Sphe3 with 1000 mg/L phenol for 24 h, which was 551 mg/L, whereas it was 105, 170, 305, 381, and 540 mg/L at Sphe3 cultures with initial phenol concentrations of 300, 500, 750, 1200, and 1500 mg/L, respectively. It was clearly shown that the phenol degradation results are in agreement with the results of growth based on OD600 as Sphe3 biomass seems to be higher up to 24 h in the presence of 1000 mg/L of phenol (Figure 1). No significant change was observed in the amount of remaining phenol in any culture after 24 h, when cells seemed to have entered the stationary phase (data not shown). In addition, no significant change in the phenol concentration was observed in control cultures with heat-killed cells, attributing the measured phenol disappearance in the Sphe3 culture to the strain’s ability to catabolize phenol.

Phenol degradation by other *Arthrobacter* strains has been reported before. Specifically, Margesin et al. reported a cold-tolerant *Arthrobacter* sp. capable of completely catabolizing 400 mg/L phenol in 72 h [42], while Karigar et al. reported that *Arthrobacter citreus* cells completely degraded 471 mg/L phenol in 24 h [17]. Similar to our results, Li et al. observed that the strain of *Arthrobacter* with Accession No. KT369868 degraded 80% of 500 mg/L phenol after 24 h [19].

### 3.2. Phenol Degradation Pathway

Aerobic degradation of phenol by bacteria varies depending on the species. Firstly, a phenol hydroxylase catalyzes the oxidation of phenol to catechol using a molecular oxygen and then catechol is degraded via the *ortho*-pathway using 1,2-catechol dioxygenase and/or via the *meta*-pathway by 2,3-catechol dioxygenase to produce a *cis, cis*-mucconic acid and/or 2-hydroxymucconic semialdehyde, respectively, which both further enter the tricarboxylic acid cycle [40].

In silico studies on the genome of *P. phenanthrenivorans* Sphe3 revealed genes likely involved in aerobic phenol catabolism. Specifically, the Sphe3 genome harbors the genes Asphe3_36590, Asphe3_35170, and Asphe3_40510 which are likely to encode the phenol hydroxylase, 1,2- and 2,3-catechol dioxygenase enzymes, respectively.

According to a BLASTP search, the Asphe3_36590 amino acid sequence shares a high identity with phenol 2-monooxygenases of other *Arthrobacter* strains (Appendix A), Asphe3_35170 shares over 91% identity with other *Arthrobacter* catechol 1,2-dioxygenases, and Asphe3_40510 shares a relatively low identity with other catechol 2,3-dioxygenases.

In order to elucidate the phenol degradation pathway in Sphe3, the transcription levels of Asphe3_36590, Asphe3_35170, and Asphe3_40510 were assessed by RT-qPCR when the strain was grown in 500 mg/L phenol. The transcription of all three genes was induced in cells grown on phenol. Specifically, Asphe3_35170 showed the highest transcriptional induction of about 125 times, while Asphe3_36590 and Asphe3_40510 showed a 38- and 77-fold change increase in mRNA expression levels, respectively, in the presence of phenol when compared to glucose (Figure 3). The induction of the aforementioned genes’ expression in the presence of phenol suggests their implication in phenol catabolism.

More evidence for the pathway utilized by strain Sphe3 for phenol degradation was obtained by the examination of phenol hydroxylase, catechol 1,2- and 2,3-dioxygenase activities in crude extracts after growth of Sphe3 cells on phenol. The cell-free extract of Sphe3 cells, grown in 500 mg/L phenol, showed a specific activity of 0.09 U mg^−1^ protein assaying phenol hydroxylase against phenol. Catechol 1,2-dioxygenase activity was measured as 0.18 U mg^−1^ protein against catechol, whereas no catechol 2,3-dioxygenase activity was detected. Similar results have been reported before for cell-free extracts from *Acinetobacter lwoffii* NL1 cells, measuring 0.13 and 1.48 U mg^−1^ of phenol hydroxylase and catechol 1,2-dioxygenase activities, respectively, while no catechol 2,3-dioxygenase activity was detected, indicating that the strain degrades phenol via the *ortho*-cleavage pathway, [5]. Additionally, in *Rhodococcus opacus*, 1CP phenol is metabolized through catechol via the *ortho*-pathway, based on catechol 1,2-dioxygenase activity [15]. Moreover, Margesin et al. reported the enzymatic activity of both catechol dioxygenases, but catechol 1,2-dioxygenase activity prevailed against catechol 2,3-dioxygenase for both mesophilic *P*. *putida* and cold-tolerant *Arthrobacter* sp. [42].

For further confirmation of the aerobic catabolic pathway of phenol in Sphe3, a rich inoculum of 3 × 10^8^ CFUs mL^−1^ free Sphe3 cells was added in MM M9 supplemented with 500 mg/L phenol under agitation at 30 °C and samples were taken at different time intervals (0, 2, 6, 12, 24, and 36 h) and analyzed by HPLC. Products from phenol transformation were identified through comparison with retention time (RT) and spectra (λ_max_) of respective standard compounds. As shown in Figure 4a, phenol (RT 6.4 min, λ_max_ 270 nm) concentration in the culture declines from 500 to 79 mg/L in only 24 h, while no phenol was detected after 36 h. During phenol transformation, the peak of catechol was identified (RT 4.7 min, λ_max_ 275 nm) at the 2 h sample and then disappeared. In addition, *cis*,*cis* muconic acid (RT 4.4 min, λ_max_ 260 nm), belonging to the *ortho*-catabolic phenol pathway, was also identified at the 2 h sample (Figure 4b), while its peak gradually became higher (Figure 4c,d). Comparable results were reported previously during phenol degradation by *Acinetobacter calcoaceticus* NCIB 8250; thus, catechol concentration increased only at the beginning, had a maximum in the first phase of phenol catabolism, and then was reduced, resulting in a complete disappearance during further phenol degradation, while *cis*,*cis*-muconic acid concentration increased at the beginning of phenol utilization but in contrast to our results, remained at low levels during further degradation [46].

Based on the gene annotation and transcription levels study, both the *ortho*- and *meta*-pathways could be active in Sphe3 for phenol degradation. However, Sphe3 seems to have an incomplete catechol *meta*-cleavage pathway. Except for the 2,3-catechol dioxygenase, no genes implicated in the *meta*-cleavage catechol pathway were found on the Sphe3 genome. The incomplete *meta*-cleavage pathway seems to validate experimental results on 2,3-dioxygenase activity and HPLC analyses, thus apart from catechol, the presence of *cis, cis* muconic acid, an intermediate of the *ortho*-pathway, was detected. Similarly, Comte et al. indicated that in *S. solfataricus* 98/2, the aromatic ring is preferentially opened through the *meta* pathway, though both degradation pathways are functional in the presence of phenol [47]. Moreover, Margesin et al. reported the enzymatic activity of both catechol dioxygenases, but catechol 1,2-dioxygenase activity prevailed against catechol 2,3-dioxygenase for both mesophilic *P*. *putida* and cold-tolerant *Arthrobacter* sp. [42]. However, in Sphe3, since transcription of Asphe3_40510 was induced in the presence of phenol but no catechol 2,3-dioxygenase activity was detected, the encoded enzyme seems to somehow be induced and/or participate in phenol catabolism but its role is yet to be determined. Different aromatic compounds and/or different concentrations can activate different metabolic pathways [41,48]. Similarly, the versatility of *Pseudomonas putida* in its genetic capability in regulating its dissimilation metabolic pathways has been demonstrated by Loh and Chua [49].

In conclusion, integrating enzyme assays results and HPLC analyses findings confirm prominence of the *ortho*-cleavage pathway for phenol metabolism in Sphe3. In addition, Sphe3 genome analysis corroborates experimental work by showing that the Sphe3 genome harbors genes encoding enzymes, such as muconolactone delta-isomerase, 3-oxoadipate enol-lactonase, and 3-oxoadipate CoA-transferase, that participate in the *ortho*-cleavage pathway of catechol as reported elsewhere [50]. Similarly, Lee et al. suggested the *ortho* pathway for phenol degradation in psychrotolerant *Arthrobacter* sp. strains, which was validated from both the bioinformatic analysis and enzyme assays of catechol 1,2-dioxygenase and catechol 2,3-dioxygenase [20]. On the contrary, Karigar et al. reported the utilization of a *meta*-cleavage pathway for phenol degradation in *Arthrobacter citreus*, based on enzyme activities of phenol hydroxylase and catechol dioxygenases combined with metabolites identification through TLC [17].

### 3.3. Phenol Degradation by Immobilized Sphe3 Cells

Since the optimal phenol degradation of Sphe3 was observed at an initial concentration of 1000 mg/L phenol, further experiments with immobilized Sphe3 cells were conducted under the same concentration. The efficiency to degrade 1000 mg phenol was evaluated in terms of assessing the remaining phenol concentration in MM M9 lacking phosphates by alginate-entrapped Sphe3 cells, with a respective bacterial inoculum, as the one used for phenol catabolism for free cells.

As shown in Figure 5, in the presence of 1000 mg/L phenol, alginate-entrapped Sphe3 cells degraded 36% of phenol after 24 h. When compared to free Sphe3 cells, immobilized cells showed a lower degradation efficiency for the first 24 h of incubation time; nevertheless, they were able to completely degrade phenol after 192 h of incubation. Such delay, compared to free cells’ activity, could be attributed to space limitation for bacterial growth inside the bead-shaped structure [51], or another possible explanation might be the lack of phosphates in the minimal medium, that likely delay bacterial growth. In contrast to free cells, alginate-entrapped Sphe3 cells were protected from phenol’s toxicity and completely degraded 1000 mg phenol after 192 h, while free cells catabolized 50% of phenol when grown in the presence of 1000 mg/L, thus entering the stationary growth phase after 24 h, as mentioned above.

Similar results have been reported before for other phenol-degrading strains. Immobilized *Bacillus* sp. SAS19 exhibited lower degradation activity compared to free bacterial cells [52], while immobilized *Pseudomonas putida* BCRC 14365 showed a slightly lower degradation rate of phenol than the free cells [44]. Moreover, immobilized *Acinetobacter* sp. strain PD12 cells exhibited a lower specific degradation rate in phenol concentrations below 300 mg/L when compared to free cells [53].

Specifically, regarding *Arthrobacter* strains, Mohanty et al. reported an improvement in phenol catabolism by immobilized *Arthrobacter* sp. regarding degradation rate and toxic compound tolerance when compared to free cells [54]. On the contrary, Karigar et al. reported no difference in the phenol degradation profile of free, alginate-, or agar-immobilized *Arthrobacter citreus* cells [17]. In particular, both free and immobilized forms of *A*. *citreus* cells were able to degrade 22 mM phenol in 8 days.

### 3.4. Phenol Catabolism—Parameters Optimization

The effects of pH and temperature parameters in the optimization of phenol catabolism in Sphe3 were investigated. Both free and immobilized Sphe3 cells retained phenol removal activity above 50% in a pH range 4–8 and exhibited optimum activity to catabolize phenol at a pH 7 value (Appendix A). For temperatures tested between 20 to 50 °C, phenol removal activity was assessed above 55% for both cell variates, while the optimum temperature was at 20 °C (Appendix A).

### 3.5. Reusability of Immobilized Cells

Considering that reusability is one of the great advantages of immobilization, but also a crucial factor for further bioremediation applications, the same batch of alginate-entrapped Sphe3 cells were tested for several cycles for phenol removal activity.

As shown in Figure 6, the immobilized cells could be efficiently reused for five reaction cycles. It can be also observed that Sphe3 beads retained more than 75% of their initial activity for five cycles of consecutive use.

At a higher number of cycles, the efficiency of the system decreased more (data not shown), probably due to cell leakage out of the beads as a result of the repetitive washes at the end of each cycle, as observed before for alginate entrapment of Sphe3 cells [35]. Lower phenol removal activity after a number of cycles could also be attributed to adsorption of possible reaction products to the immobilization matrix, which subsequently alters its reactive mechanical stability, therefore reducing the catalytic activity of the involved enzymes [36].

Similarly, many previous studies have reported a stability or a slight decrease in phenol degradation efficiency when increasing cycles of reuse. The *Debaryomyces* sp. strain entrapped in Ca-alginate beads containing nano-Fe_3_O maintained its original phenol-degrading efficiency after 10 cycles of repeated batch operations [55]. Basak et al. reported that after five times of reuse, phenol degradation efficiency decreased gradually to 78% for *Candida tropicalis* PHB cells immobilized to alginate beads and after ten cycles, efficiency dropped to 3.74% [51], while Chris Felshia et al. reported a 40% efficiency in phenol removal after five-cycle reuse of the encapsulated-in-whey-protein strain *Bacillus lichenformis* SL10 [16]. The successful use of Sphe3 alginate beads up to five cycles is of great importance, since it could significantly reduce the operation cost in potential industrial applications.

### 3.6. Phenol Degradation Efficiency by Immobilized Cells after 30 Days of Storage

An important consideration on how the immobilized cells could be used in future bioremediation applications is the maintenance of their phenol removal efficiency upon storage. Thus, the alginate-entrapped *Ps. phenanthrenivorans* Sphe3 cells were stored for 10, 20, and 30 days at 4 °C and then introduced to medium containing 1000 mg/L phenol. Phenol removal efficiency was determined after 48 h incubation. It was observed that after 30 days, the stored bacterial alginate beads retained more than 70% of their initial phenol degradation activity (Figure 7).

Nandy et al. observed that the encapsulated *Pseudomonas oleovorans* ICTN13 cells stored at 4 °C for 30 days could reduce phenol concentration by 53% in 24 h, retaining half of their catalytic activity, whereas longer storage time (beyond 30 days) reduced phenol removal activity significantly [11]. Moreover, Banerjee and Ghoshal observed no reduction in phenol degradation efficiency by alginate beads with immobilized *Bacillus cereus* AKG1 MTCC9817 and AKG2 MTCC 9818 cells upon 30 days of storage [56].

## 4. Conclusions

*Pseudarthrobacter phenanthrenivorans* Sphe3 is capable of efficiently catabolizing phenol as the sole source of carbon and energy, mainly via the catechol *ortho*-cleavage route. The main degradation pathway was elucidated by transcription analysis of genes involved and the enzyme activity of their corresponding enzymes as well as by metabolic products detection of phenol degradation by HPLC analysis.

The immobilization of Sphe3 cells in alginate seems to protect the cells from the toxic compound, as a complete removal of 1000 mg phenol/L was observed at 192 h, whereas while free cells exhibited higher catabolic efficiency during the first 24 h, they failed to catabolize more than 50% subsequently. In addition, alginate-entrapped cells were reused for five cycles and stored for a month retaining their phenol removal activity over 75% and 70%, respectively.

Considering the reusability and stability of immobilized cells upon storage are crucial coefficients for their applicability in a bioremediation system; *P*. *phenanthrenivorans* Sphe3 is proved to meet these requirements making it a fine candidate for phenol removal.

## Figures and Tables

**Figure 1 microorganisms-11-00524-f001:**
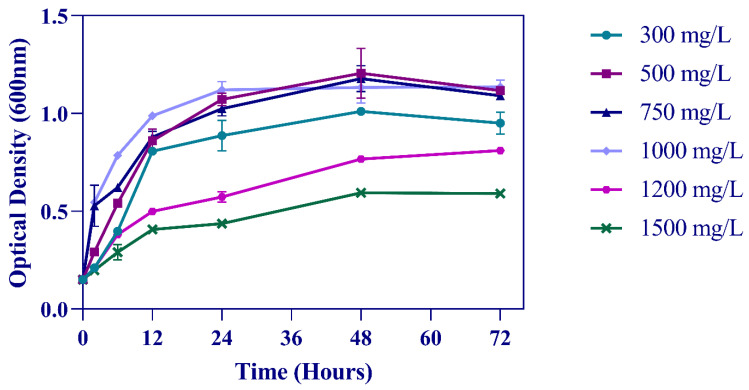
Sphe3 growth curves in different phenol concentrations. Solid lines with circle, square, triangle, rhombus, polygon, and ex symbols represent measurements of Sphe3 growth in the presence of 300, 500, 750, 1000, 1200, and 1500 mg/L phenol, respectively. Error bars indicate standard deviation of three measurements.

**Figure 2 microorganisms-11-00524-f002:**
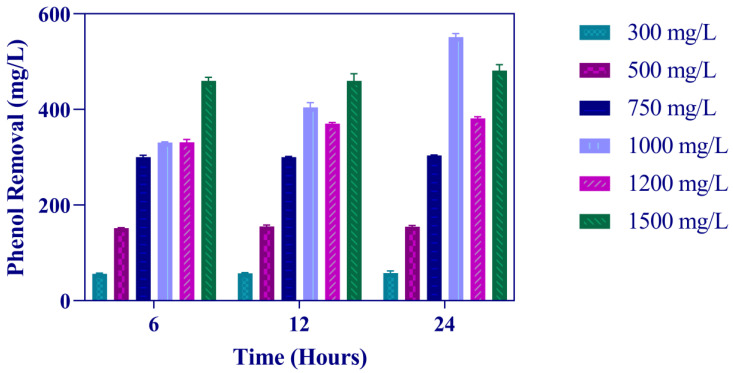
Phenol removal by Sphe3 cultivated in the presence of 300–1500 mg/L phenol, at 30 °C, after 6, 12, and 24 h. Error bars indicate standard deviation of three measurements.

**Figure 3 microorganisms-11-00524-f003:**
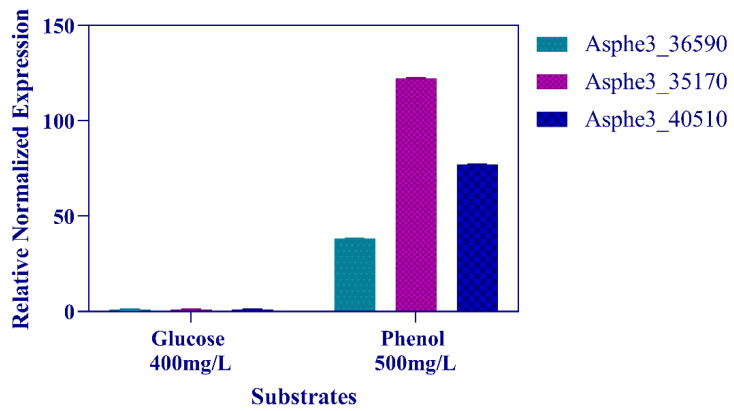
Asphe3_36590, Asphe3_35170, and Asphe3_40510 transcription quantification monitored by RT-qPCR in Sphe3 grown on glucose and phenol, each as the sole carbon and energy source. Values represent the mean relative gene expression normalized to the housekeeping gene *gyr*β ± standard deviations of three individual replicates. Gene expression levels in glucose were used as a calibrator.

**Figure 4 microorganisms-11-00524-f004:**
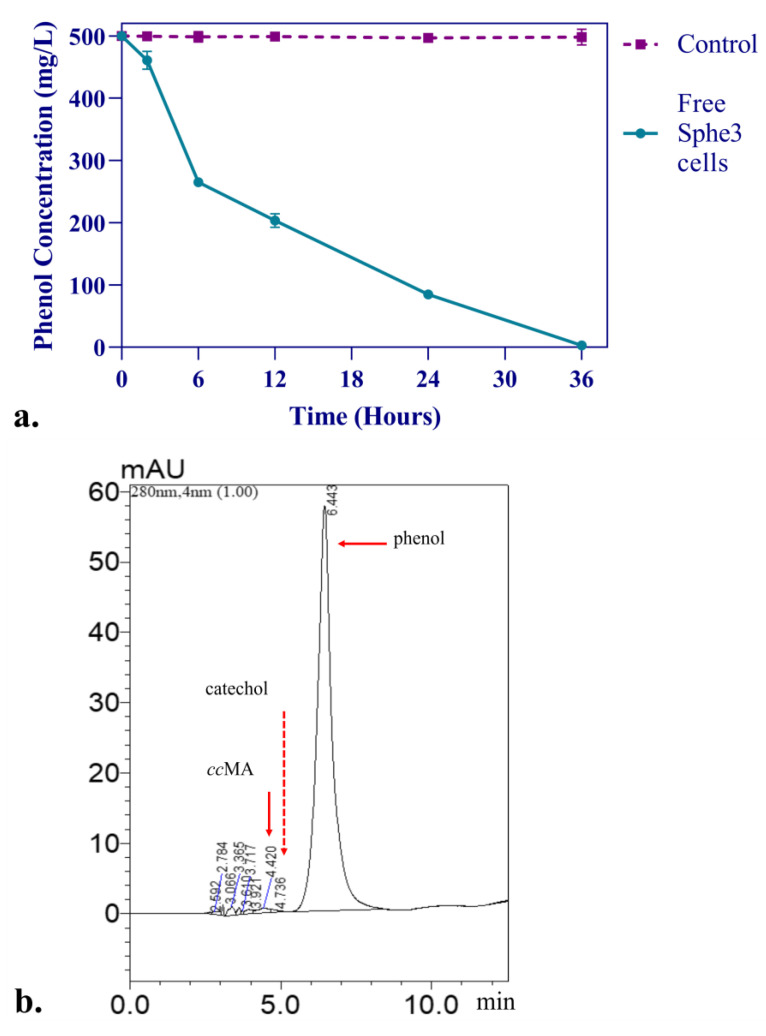
The profile of phenol concentration changes in free Sphe3 cell culture at different timepoints. Error bars indicate standard deviation of three measurements (**a**). HPLC analysis of sample taken from Sphe3 culture in phenol at 2 h (**b**). HPLC analysis of sample taken from Sphe3 culture in phenol at 12 h (**c**) and at 24 h (**d**). Red arrows indicate *cc*MA, catechol, and phenol.

**Figure 5 microorganisms-11-00524-f005:**
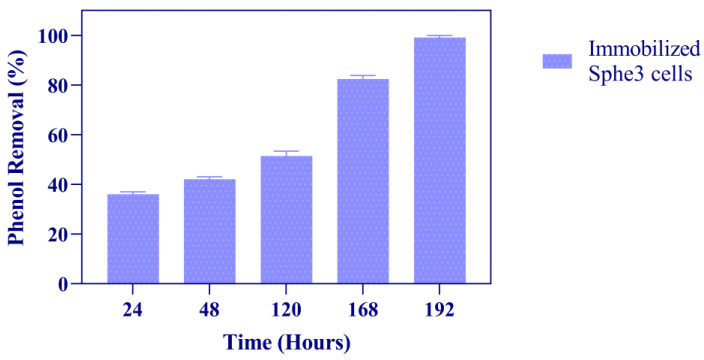
Phenol degradation by immobilized *Ps*. *phenanthrenivorans* Sphe3 in the presence of 1000 mg/L phenol, at 30 °C, expressed as % phenol removal activity after 24, 48, 120, 168, and 192 h. Error bars indicate standard deviation of three measurements.

**Figure 6 microorganisms-11-00524-f006:**
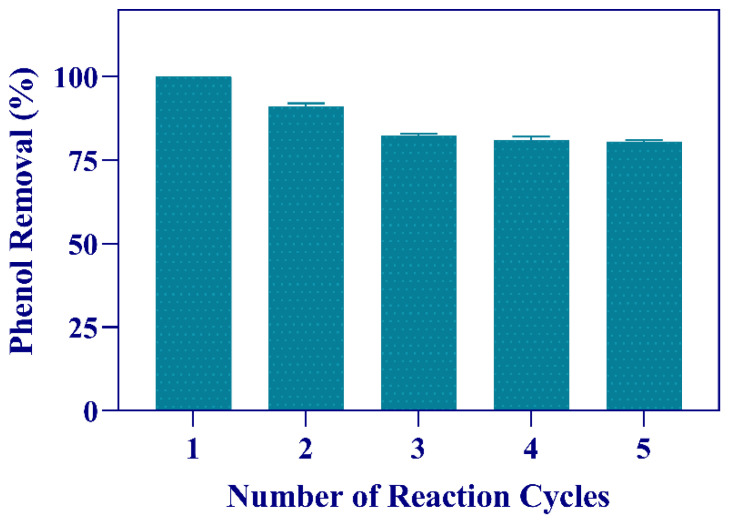
Reusability of immobilized Sphe3 cells for phenol removal. The activity of the first cycle was considered as 100%. Each reaction cycle was carried out for 48 h. Error bars indicate standard deviation of three measurements.

**Figure 7 microorganisms-11-00524-f007:**
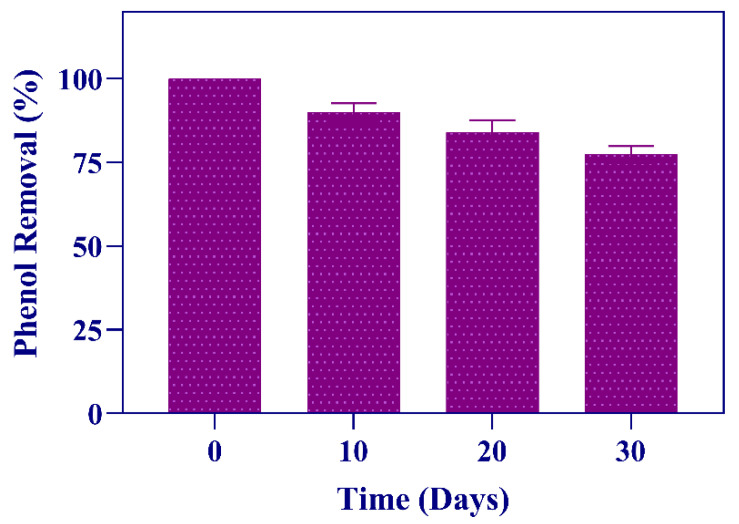
Phenol removal activity of immobilized Sphe3 cells upon storage at 4 °C for 1 month. The activity on the first day was set to 100%. Each reaction cycle was carried out for 48 h. Error bars indicate standard deviation of three measurements.

## Data Availability

All data generated or analyzed during this study are included in this published article (and its Appendix A).

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
