# Peer review of "Phenol Degradation by Pseudarthrobacter phenanthrenivorans Sphe3"

_microorganisms, 2023, doi:10.3390/microorganisms11020524_

Round 1
Reviewer 1 Report
The manuscript microorganisms-2218415 presents the results of the study conducted with an aim to investigate phenol degradation by Pseudarthrobacter phenanthrenivorans Sphe3 cells and the pathway used by the bacterium for phenol catabolism.
The study design is acceptable, methods are well described and obtained results are clearly presented and discussed.
I recommend acceptance of the paper "Phenol degradation by Pseudarthrobacter phenanthrenivorans Sphe3" in its present form.
Reviewer 2 Report
The article entitled "Phenol degradation by Pseudarthrobacter phenanthrenivorans Sphe3" is an article describing the ability of phenol degradation by a strain of Pseudarthrobacter.
The main drawback of this manuscript is the lack of statistical analysis of obtained results. The Authors showed that the strain is able the degrade phenol but statistical analysis is need to proof which conditions are the best/optimal. In many figures I can see error bars, so the statistic could be done.
Also figure 2 could be missunderstood. I will suggest to change the unnits of phenol removal from % to the concentration, as now it is hard to calculate (looking only on the picture) how many mg were degraded. The concentration tested in this experiment ranged from 300 to 1500 mg/l. It is quite wide range. 50% of 300 mg/l is something very different from 50% of the highest concentration (it is just the example to visualize the problem).
In my opinion without statistical analysis the manuscript can not be published, especially in Microorganisms.
Reviewer 3 Report
Authors presented an evaluation of the phenol degradation capacity of Pseudarthrobacter phenanthrenivorans, because of the enzymatic activity, HPLC products and gene expression, they conclude that this microorganism degrade phenol by the catechol-Ortho pathway. Nevertheless, some experimental results are not clear.
In Figure 1, why the growth was lower at 300 mg/L of phenol than 500, 750 and 1000 mg/L?, the lowest growth was observed at 1500 mg/L. What is the explanation of this behavior?.
If growth and phenol removal was higher at 1000 mg/L, why this concentration was not used for Asphe 3…genes expression?
Round 2
Reviewer 2 Report
Now the article could be accepted for publishing in MIcroorganisms.
Reviewer 3 Report
The reviewers requests were answered.